# Named Entity layer in Estonian UD treebanks

**Kadri Muischnek**
University of Tartu
`kadri.muischnek@ut.ee`

**Kaili Müürisep**
University of Tartu
`kaili.muurisep@ut.ee`

## Abstract

In this paper we will introduce two new language resources, two NE-annotated corpora for Estonian: Estonian Universal Dependencies Treebank (EDT, 440,000 tokens) and Estonian Universal Dependencies Web Treebank (EWT, 90,000 tokens). Together they make up the largest publicly available Estonian named entity gold annotation dataset. Eight NE categories are manually annotated in this dataset, and the fact that it is also annotated for lemma, POS, morphological features and dependency syntactic relations, makes it more valuable. We will also show that dividing the set of named entities into clear-cut categories is not always easy.

## 1  Introduction

Named entity recognition (NER) is an important sub-task of information extraction. In order to build a NER tagger, one first needs to annotate a corpus for named entities (NE). In this paper we introduce two NE-annotated corpora for Estonian: Estonian Universal Dependencies Treebank[1] (EDT) and Estonian Universal Dependencies Web Treebank[2] (EWT). By annotating these two resources for NE, we have aimed at broad coverage of genres, writing styles and correct vs relaxed compliance to the Estonian spelling rules.

Although there are previous NE-annotated resources for Estonian, we regard enriching existing UD corpora with NE annotation an important effort as UD annotations can support both manual annotation and help to build better NER models. We were also encouraged by the reports on similar efforts for Finnish (Luoma et al., 2020),

Danish (Hvingelby et al., 2020) and Norwegian (Jørgensen et al., 2020).

In this paper, we first describe the underlying Estonian UD treebanks in Section 2.1. Section 2.2 introduces the NE categories that are distinguished in the dataset and discusses some gray areas between these classes. Corpus statistics is presented in Section 2.3 and a brief overview of related work is given in Section 3.

The NE annotations are included in the release 2.12 of Estonian UD treebanks.

## 2  Corpus and annotations

### 2.1  Estonian UD treebanks

Universal Dependencies[3] (De Marneffe et al., 2021) is an open community effort for annotating dependency treebanks using consistent annotation scheme for different human languages. Currently UD treebank collection entails nearly 200 treebanks in over 100 languages.

There are two Estonian UD treebanks: Estonian Universal Dependencies Treebank EDT and Estonian Universal Dependencies Web Treebank EWT. EDT contains ca 440,000 tokens in ca 30,000 sentences and its texts cover three central text types of normed written language: fiction, journalism and scientific writing. The text types of the treebank are not balanced: journalism with ca 270,000 tokens makes up more than half of the treebank, whereas fiction (ca 68,000 tokens) and scientific texts (ca 95,000 tokens) comprise the other half. EWT consists of texts from blog posts, online comments and discussion forums, it contains ca 90,000 tokens in ca 7000 sentences.

Universal Dependencies annotation is described thoroughly on their website[4]. For the task of NE annotation it is relevant to point out that there is a special POS-tag for proper nouns (PROPN) and a

---

[1]https://universaldependencies.org/treebanks/et_edt/
[2]https://universaldependencies.org/treebanks/et_ewt/

[3]https://universaldependencies.org/
[4]https://universaldependencies.org/guidelines.html

special syntactic relation 'flat', that is used for ex-ocentric (headless) structures, also for multiword names. So it is relatively easy to pre-annotate the majority of NEs automatically, using these UD annotations. However, there are still some NEs that don't include a proper noun, so an annotator still has to go through the entire text carefully. Also, the exact extent of a named entity and its category have to be marked manually.

## 2.2 NE categories and annotation scope

Martin and Jurafsky (2021) summarize the common practice for NE annotation, noting that although a named entity is anything that can be referred to with a proper name; often also dates, times, and other kinds of temporal expressions, and even numerical expressions like prices are also included while annotating and tagging NE-s.

In our project we, at least for the time being, have annotated only "proper" NE-s, i.e. the entities that contain a proper noun or otherwise refer to a specific object like a title of a book, a film, a song etc. We have classified these entities into eight categories: persons `Per`, locations `Loc`, geo-political entities `Gep`, organizations `Org`, products `Prod`, events `Eve`, NE-s that do not fit into aforementioned categories (`Other`) and NE-s that can't be categorized due to the lack of information (`Unk`).

Often a proper noun or a title is accompanied by a headword indicating the type of the NE and thus providing valuable information. In Estonian writing, these headwords are not capitalized and they can both follow or precede the proper noun, e.g. *Tartu linn* 'Tartu city' or *romaan "Sõda ja rahu"* 'novel "War and Peace"'. Headwords of named entities are included in the annotation span, but personal titles like *härra Kask* 'mister Kask' are not.

The texts of EDT originate from the period 1998—2007 and the capitalization conventions have changed slightly during this period. *Internet* and *Sudoku* are among examples of unstable capitalization, they tend to be capitalized more in the earlier texts. Also, names of "newer" diseases like *Ebola* or *Covid* tend to be capitalized, although the language specialists suggest that a lower-case versions should be used. Of course a named entity should be annotated as such, regardless of whether it meets the spelling standard or not. On the other hand, capitalization in written Estonian is a signal that the word is a proper noun and if the text more or less follows the norms of the written language, POS tagging relies on capitalization while making the distinction between common and proper nouns. So *Internet* is a proper noun and *internet* a common noun. NE annotation, in turn, relies on POS tags, so *Internet* is a NE and *internet* is not.

Similarly, names of celestial bodies like *Maa* 'Earth' or *Kuu* 'Moon' are capitalized if referring to "a certain place in the Universe" and are treated as named entities there.

In the Estonian Web Language Treebank EWT, the texts differ from each other as to whether the author follows the norms of written language or, deliberately, does not care about them. Some writers do not use capitalization at all, others use it in an inconsistent manner. So, the POS tagging in EWT can't rely so much on capitalization but the annotator has to understand whether the reference is unique or not and NE annotations and POS tags still need to be consistent with each other.

While dividing the set of NE-s into types or categories we have put more emphasis on consistency (similar entities have to be grouped together) than on "absolute justness". So, in case that the annotators pointed out that they are persistently confused about making a clear-cut distinction between certain categories, we considered re-drawing the line. An example of `Loc` and `Gep` will be presented hereinafter.

We will now present our categories one by one.

The category `Per` includes, in addition to person names, also names of animals and imaginary creatures. Family names are annotated as `Per` even though they refer to several people, e.g. *perekond Tamm* 'Tamm family'. In internet forums, usernames are annotated as `Per`, but they are quite different from person names in general, so may be it would be a good idea to annotate them as examples of a subtype of `Per`.

The category `Loc` includes names of landscape objects like rivers or hills, and also names of man-made landscape objects like roads or settlements.

Geo-political entities `Gep` are entities that originally stand for locations, but are often represented in texts as agents – they can decide or say something etc. It is a typical case of metonymy: state or city is seen as the incarnation of its people or its governing body. This category was introduced in the annotation scheme of the Automatic Content Extraction program (ACE) (Mitchell et al.,

|       | news | fiction | sci  | other | ewt  |
|-------|------|---------|------|-------|------|
| Per   | 5718 | 1202    | 1100 | 432   | 1896 |
| Loc   | 2498 | 305     | 445  | 63    | 268  |
| Gep   | 3324 | 230     | 442  | 42    | 318  |
| Org   | 2578 | 47      | 300  | 73    | 320  |
| Prod  | 1588 | 88      | 401  | 8     | 819  |
| Event | 320  | 5       | 61   | 1     | 51   |
| Other | 22   | 1       | 2    | 0     | 9    |
| Unk   | 33   | 6       | 9    | 0     | 5    |

Table 1: Counts of named entities in treebanks

|       | news | fiction | sci  | other | ewt  |
|-------|------|---------|------|-------|------|
| Per   | 2.79 | 1.77    | 1.31 | 4.70  | 2.09 |
| Loc   | 1.22 | 0.45    | 0.53 | 0.68  | 0.30 |
| Gep   | 1.62 | 0.34    | 0.52 | 0.46  | 0.35 |
| Org   | 1.26 | 0.07    | 0.36 | 0.79  | 0.35 |
| Prod  | 0.77 | 0.13    | 0.48 | 0.09  | 0.90 |
| Event | 0.16 | 0.01    | 0.07 | 0.01  | 0.06 |
| Other | 0.01 | 0.00    | 0.00 | 0.00  | 0.01 |
| Unk   | 0.02 | 0.01    | 0.01 | 0.00  | 0.01 |

Table 2: Counts of named entities in percentile points

2003). Categorizing named entities as `Loc` or `Gep` in a consistent manner turned out to be a difficult task for the annotators, so, remaining true to our principle of prioritizing annotation consistency, we made a simplifying decision that a name of a state is always an example of `Gep`, whereas a name of a city or other settlement can be annotated as `Loc` or `Gep` depending on the context.

The decision to annotate all state names as geopolitical entities can be seen as an oversimplification, but our annotators pointed out that they kept doubting about the correct label especially in this case. Even if the word denoting a state is in a spatial case form, it is not a firm proof that it has spatial meaning and should be annotated as a place. For example, in a sentence *Raha jõudis Eestisse anonüümselt.* 'The money arrived in Estonia anonymously.' one can't infer from the text whether *Eestisse* 'in Estonia' here means the Estonian land or the Estonian state, the economic space governed by Estonian legislation.

The category `Org` is relatively straightforward. Yet there exists a grey area between organizations and products produced by those organizations. For example, the name of a newspaper can stand both for an issue of a newspaper, e.g. *in the latest Ekspress an article about elections was published* and for the editorial board of this newspaper, e.g. *Ekspress's view on elections is presented in this article*.

The category `Prod` includes man-made objects, also abstract entities such as ideas or theories. Again, the category seems to be easy at first glance, but depending on a context, a product can be presented as a location in texts: a person is in a building, a cat is in a cupboard, a fly is in a bowl. Also, products have a certain overlap with events. A movie is a product, but what about a theatre performance, taking place on a certain time and in a

certain place? There is also a gray area of buildings and other man-made landscape objects, e.g. airports.

So, we have seen a few times that there exist grey areas at the borders between NE categories and, perhaps from the semantic point of view also other intersectional categories besides `Gep` would be justified. But the main objective of our work is to build a NER tagger and having too many too small NE types would hamper the NER task.

The category `Other` is used to annotate NEs that do not fit into aforementioned categories, the examples include *U3 projekt* 'the U3 project' or *Dow indeks* 'The Dow Index'. As seen from Table 1, it is the rarest of the NE categories.

The category `Unk` is used for annotating NEs which meaning is not clear. This category is more frequent in web texts, although, compared to other NE categories, it is infrequent there also. A good example of an unknown named entity originates from a fiction text describing a non-sensical lecture about *blanko-idosseeritud Pardakonossement*. Both words do not exist in Estonian, but it can be inferred from the context that *blanko-idosseeritud* is a past participle and *Pardakonossement* is a proper noun.

## 2.3 Annotation process

At the beginning of the project, it was clear that there was a need to annotate the entire Estonian UD treebank (approximately 530,000 tokens) in a consistent manner and also keep in mind that our created annotation should not differ drastically from the previous named entity annotation efforts.

The EDT treebank was pre-annotated automatically, based on name lists primarily including frequent person names. With the help of syntactic annotations, the extent of the named entity was at-

```
11  kus       kus        ADV    D                               12  advmod  12:advmod   _
12  elab      elama      VERB   V    Mood=Ind|Number=Sing       9   acl     9:acl
13  Eesti     Eesti      PROPN  S    Case=Gen|Number=Sing       14  nmod    14:nmod     NE=B-Gep
14  juurtega  juur       NOUN   S    Case=Com|Number=Plur       15  nmod    15:nmod
15  hokimehe  hoki_mees  NOUN   S    Case=Gen|Number=Sing       21  nmod    21:nmod     _
16  Håkan     Håkan      PROPN  S    Case=Nom|Number=Sing       15  appos   15:appos    NE=B-Per
17  Loobi     Loop       PROPN  S    Case=Gen|Number=Sing       16  flat    16:flat     NE=I-Per
18  Kihnu     Kihnu      PROPN  S    Case=Gen|Number=Sing       19  nmod    19:nmod     NE=B-Loc
19  saarelt   saar       NOUN   S    Case=Abl|Number=Sing       20  obl     20:obl      NE=I-Loc
20  pärit     pärit      ADV    D                               21  advmod  21:advmod   _
21  isa       isa        NOUN   S    Case=Nom|Number=Sing       12  nsubj   12:nsubj    _
22  Paul      Paul       PROPN  S    Case=Nom|Number=Sing       21  appos   21:appos    NE=B-Per
```

Figure 1: Corpus example: annotated clause *where hockey player with Estonian roots Hakan Loob's father Paul from Kihnu island lives*

tempted to be identified, and for remaining proper names, annotations `B-Unk` (first member of the named entity) and `I-Unk` (subsequent members of the named entity) were added.

Initially, there were 3 student annotators who annotated the texts; at the first stage texts were annotated by two students, the annotations compared and the discrepancies solved. The students had different skills and availability, so eventually, one student continued to work alone. If the annotator felt that the solution was not unambiguous, he wrote a question into the log-file, which was later discussed with supervisors. Lists of annotated named entities were also compiled and reviewed together. The EWT corpus, which is smaller in size but more complex in content, was annotated by a student and then checked and corrected by supervisors. This method for annotation does not allow for calculations to assess the inter-annotator agreement measures but we believe that a multi-person, multiple-check annotated corpus is the best that could be created given limited resources.

## 2.4 Corpus statistics

Tables 1 and 2 show the raw and normalized NE frequencies in EDT and EWT and the distribution of NEs in different text types. EDT contains the main text classes of normed written language: newspaper texts, fiction and scientific texts. Only one text, containing example sentences from a scientific work about Estonian valency patterns, plus sentences from different news texts, belongs to the text class "other". EWT contains the text classes of user-generated content: blog posts, comments and forum texts.

In EDT, the frequencies are distributed as could be expected: newspaper texts have the highest density of NEs, fiction texts contain lot of person

names. Scientific texts include references, that, somewhat unnaturally, increase the frequency of person names in them. The text class "other" does not represent normal text: the example sentences of the valency frames include person names (never pronouns or common nouns referring to a human) wherever a word denoting a human was possible, e.g. *Mary saw John*.

EWT forum texts include usernames, that are annotated as `Per` and the users also address each other using their usernames. In web forums people also discuss and rate various products, which raises the frequency of `Prod` category.

## 3 Related work

### 3.1 Estonian NE-annotated corpora

There are two previous NE-annotated corpora for present-day Estonian and one for historical Estonian. Tkachenko and colleagues (Tkachenko et al., 2013) have annotated four NE categories (persons, locations, organizations and other) in a 185,000-token dataset.

New Estonian NER dataset[5] contains ca 140,000 tokens and the annotated NEs are divided into 7 categories: persons, organizations, locations, geo-political entities, titles, products and events. In addition to "proper NEs", also dates, times, percents and currencies are annotated. During this project also Tkatchenko's dataset was re-annotated. The resulting datasets use hierarchical annotation, which we regard useful, but for the time being have refrained from using it in order to make the task easier for the annotators.

In a corpus of historical Estonian, a collection of parish court records from the 19th century (Orasmaa et al., 2022) seven NE categories are annotated: person, location, organization, location-

---

[5]https://github.com/TartuNLP/EstNER_new

organization, artefact, other and unknown. The parish court records make up a text type of its own, but their NE typology is similar to that of our corpora; the category 'location-organization' is essentially the same as our `Gep` and the category 'artefact' is similar to our `Prod`.

### 3.2 Other NE-annotated resources based on UD annotations

In the Finnish corpus (Luoma et al., 2020), six NE categories have been annotated: person, organization, location, product and event names as well as dates. In order to to avoid ambiguity-creating categories, geopolitical entities are annotated as locations, but the authors admit that for applications where the resolution of the ambiguity is not critical, there may be merit to the adoption of possibly ambiguity-creating type like geo-political entity.

In the Danish NE-annotated corpus (Hvingelby et al., 2020) four NE classes are annotated: location, organisation, person and miscellaneous, following the guidelines of the CoNLL-2003 NE annotation scheme (Sang and Meulder, 2003). They also report that it was difficult for the annotators to distinguish between locations and organizations in certain cases.

In the Norwegian UD treebank (Jørgensen et al., 2020) the categories of person, organization, location, geo-political entity, product and event have been annotated. Geo-political entities are subcategorized as either GPE with a locative sense or GPE with an organization sense. However, while annotating the corpus with those categories, annotators had some difficulties with making the distinction between the subcategories of the GPE entity types. Building on the experience of NOrNE annotation effort, we did not attempt at dividing the category of `Gep` into subcategories.

### 4 Conclusions and future directions

We have presented two manually annotated NER datasets for Estonian. The annotated texts represent the core text types of normed written language as well as several text types of the user-generated content of the web. The annotated NEs fall into eight categories: persons, locations, geo-political entities, organizations, products, events, other NEs that can't be classified into aforementioned categories and NEs of unknown category. The category of geo-political entities is a hybrid category between location and organization. Although we

noticed that there exist also other cases of systematic metonymy besides using a location name to note the people connected with this location, we did not introduce more NE types as we did not want to divide the NEs into too many too small categories.

Obviously the next step would be building NER models using this dataset. Also, as the web treebank EWT is being developed further, annotating new genres of web texts with UD annotations, we plan to add the new texts into our dataset.

### Acknowledgments

This work has been supported by the Estonian National Programme "Estonian Language Technology" via grants EKTB7 "Estonian Universal Syntax: Resources and Applications" and EKTB75 "Basic resources for semantic analysis".

We would like to thank Mihkel Rünkla and other annotators.

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
