# OpenReview forum: "Named Entity layer in Estonian UD treebanks"
_NoDaLiDa/2023/Conference — NoDaLiDa 2023_

### Official Review · Reviewer_mTWQ · 2023-03-08
**Valuable new annotated data sets for Estonian; some issues to discuss.**

**Rating:** 7
**Confidence:** 4

**Review:**

This paper presents two new annotated data sets for Estonian, consisting of two pre-existing UD treebanks that have in addition been annotated for named entities. The authors discuss some of the issues involved in delimiting different NE categories (for example, locations vs. geopolitical entities) and describe how they have adjusted the annotation guidelines to promote consistency. The resulting data sets are described quantitatively.

These data sets are clearly valuable resources for Estonian and as such worth publishing. However, I do not find the discussion of the annotation principles completely convincing in all respects, as specified below. In addition, the paper could be strengthened by including information about inter-annotator agreement, perhaps also reporting an experimental NER evaluation based on the new resources (although this might be asking too much in a short paper).

Specific comments:

If I understand correctly, a phrase can only be considered a named entity if it contains at least one word tagged PROPN in the UD scheme. This strikes me as a constraint that can lead to an unsystematic treatment of certain types of entities, such as movie titles. If my assumption is correct, then it seems to follow that “Citizen Kane” is a named entity but “The godfather” is not. Is this really correct and (if so) is it desirable? Some examples in the paper also suggests that containing a word tagged PROPN is not only necessary but also sufficient. For example, treating (the equivalent) of “Achilles tendon” as a named entity simply because “Achilles” is a proper name does not seem right to me.

I also find some of the heuristics used to improve consistency less convincing. For example, deterministically annotating state names as geopolitical entities and cities as locations seems to be a case of trading validity for reliability, since the distribution of the tags GEP and LOC in the data will not correctly reflect the (original) semantic content of the categories.


**Paper Type:**

Short paper

---

### Official Review · Reviewer_PBoU · 2023-03-10
**Well written paper about the annotation of the largest publicly available Estonian NE gold annotation dataset.**

**Rating:** 6
**Confidence:** 3

**Review:**

The authors follow similar efforts for Scandinavian and don't introduce any novelty in the annotation of NE.
According to capitalisation and UD annotation authors manually annotate only "proper" NEs of only eight categories. This is significant effort for Estonian, however, I am nor sure if it is state-of-the-art data annotation. If this paper will be included in the conference program authors should clarify the process of the annotation (annotators involvement, annotation counts, speed, quality comparison etc.).

**Paper Type:**

Short paper

---

### Decision · Program_Chairs · 2023-03-17

Accept